# Attempting to Create a Pathway to 15-Deacetylcalonectrin with Limited Accumulation in Cultures of *Fusarium Tri3* Mutants: Insight into Trichothecene Biosynthesis Machinery

**DOI:** 10.3390/ijms25126414

**Published:** 2024-06-11

**Authors:** Ena Kasahara, Yuna Kitamura, Miho Katada, Masashi Mizuki, Natsuki Okumura, Tomomi Sano, Yoshiaki Koizumi, Kazuyuki Maeda, Naoko Takahashi-Ando, Makoto Kimura, Yuichi Nakajima

**Affiliations:** 1Graduate School of Bioagricultural Sciences, Nagoya University, Furo-cho, Chikusa-ku, Nagoya 464-8601, Aichi, Japan; kasahara.ena.t0@s.mail.nagoya-u.ac.jp (E.K.); kitamura.yuna.s0@s.mail.nagoya-u.ac.jp (Y.K.); mhkatada@gmail.com (M.K.); miyaviii0701@icloud.com (M.M.); natsukinoinoi@gmail.com (N.O.); tmm_sn@yahoo.co.jp (T.S.); kmaeda@agr.nagoya-u.ac.jp (K.M.); 2Graduate School of Science and Engineering, Toyo University, 2100 Kujirai, Kawagoe 350-8585, Saitama, Japan; s46d02200015@toyo.jp (Y.K.); ando_n@toyo.jp (N.T.-A.)

**Keywords:** 15-*O*-acetyltransferase encoded by *Tri3*, calonectrin (CAL)-specific C-15 deacetylase activity, *Fusarium commune*, trichothecene pathway intermediate 15-deacetylcalonectrin (15-deCAL), *Tri1* disruption mutant

## Abstract

The compound 15-deacetylcalonectrin (15-deCAL) is a common pathway intermediate in the biosynthesis of *Fusarium* trichothecenes. This tricyclic intermediate is metabolized to calonectrin (CAL) by trichothecene 15-*O*-acetyltransferase encoded by *Tri3*. Unlike other trichothecene pathway *Tri* gene mutants, the Δ*tri3* mutant produces lower amounts of the knocked-out enzyme’s substrate 15-deCAL, and instead, accumulates higher quantities of earlier bicyclic intermediate and shunt metabolites. Furthermore, evolutionary studies suggest that *Tri3* may play a role in shaping the chemotypes of trichothecene-producing *Fusarium* strains. To better understand the functional role of Tri3p in biosynthesis and evolution, we aimed to develop a method to produce 15-deCAL by using transgenic *Fusarium graminearum* strains derived from a trichothecene overproducer. Unfortunately, introducing mutant *Tri3*, encoding a catalytically impaired but structurally intact acetylase, did not improve the low 15-deCAL production level of the Δ*Fgtri3* deletion strain, and the bicyclic products continued to accumulate as the major metabolites of the active-site mutant. These findings are discussed in light of the enzyme responsible for 15-deCAL production in trichothecene biosynthesis machinery. To efficiently produce 15-deCAL, we tested an alternative strategy of using a CAL-overproducing transformant. By feeding a crude CAL extract to a *Fusarium commune* strain that was isolated in this study and capable of specifically deacetylating C-15 acetyl, 15-deCAL was efficiently recovered. The substrate produced in this manner can be used for kinetic investigations of this enzyme and its possible role in chemotype diversification.

## 1. Introduction

*Fusarium sporotrichioides* and *Fusarium graminearum* species complex (*Fg*) are known to produce trichothecenes oxygenated at C-3 and C-15 [1,2,3,4]. During biosynthesis of diverse groups of *Fusarium* trichothecenes, the second cyclization of isotrichotriol to isotrichodermol (ITDmol) proceeds non-enzymatically. ITDmol is then acetylated at C-3 to yield isotrichodermin (ITD) [5], followed by C-15 hydroxylation to produce 15-deacetylcalonectrin (15-deCAL) [6,7,8,9,10]. Subsequently, an enzyme encoded by *Tri3* (Tri3p) accepts 15-deCAL as a substrate and acetylates the hydroxy group at C-15. The Tri3p enzyme has recently been hypothesized to move this tricyclic intermediate outside of the toxisome membrane as its reaction product calonectrin (CAL) [11,12]. After CAL, the biosynthetic pathway diverges depending on the type of trichothecenes [13], namely, type A trichothecenes (without a ketone at C-8) [14], such as T-2 toxin produced by *Fusarium sporotrichioides*, and type B trichothecenes (with a ketone at C-8) [14], such as nivalenol (NIV) and deoxynivalenol (DON) produced by *F. graminearum* (Figure 1). The major *Fusarium* trichothecene product that accumulates in cereal grains is DON [15,16], whose dietary exposure causes health problems, such as diarrhea, vomiting, gastrointestinal hemorrhage, and inflammatory responses in humans and animals [17,18].

DON is often co-contaminated with a relatively stable last intermediate in the biosynthetic pathway, either 3-acetyldeoxynivalenol (3-ADON) or 15-acetyldeoxynivalenol (15-ADON), depending on the chemotype of the *F. graminearum* strain (Figure 1) [19]. Interestingly, the rate of amino acid substitution in *FgTri3* of the 3-ADON chemotype is significantly greater than that of 15-ADON and NIV chemotypes [20]. Such chemotype-specific shifts in functional constraints suggest the involvement of *FgTri3*-encoded C-15 acetylase (FgTri3p) in chemotype diversification. However, functional studies demonstrated that the determinant differentiating the 3-ADON and 15-ADON chemotypes is the *Tri8*-encoded trichothecene deacetylase (Figure 1), which hydrolyzes C-15 and C-3 acetyl groups, respectively, of 3,15-diacetyldeoxynivalenol (3,15-diADON) [21,22].

Given the importance of *Tri8* in chemotype diversification, a question arises regarding the role of *FgTri3* in this process: does C-15 acetylation activity toward DON also play a role in determining chemotypes to some extent? If 3-ADON chemotype’s FgTri3p is highly specific to 15-deCAL and exhibits much weaker activity toward DON and 3-ADON in comparison to other chemotypes, the non-occurrence of 15-ADON can be reasonably explained. Conversely, if FgTri3p of the 15-ADON chemotype shows significantly greater catalytic activity toward these trichothecenes than 3-ADON chemotype’s enzyme, the persistence of 15-ADON is more likely (Figure 1). To gain insight into this question, it is crucial to kinetically characterize the FgTri3p acetylase of different chemotypes in detail using 15-deCAL, DON, and 3-ADON as substrates. The latter two trichothecenes can easily be isolated in large quantities from the solid and liquid culture, respectively, of the 3-ADON chemotype [22,23,24]. However, *Fusarium* mutants blocked at the C-15 acetylation step, obtained either by ultraviolet (UV) irradiation or *Tri3* disruption, accumulated considerable amounts of earlier bicyclic and shunt metabolites (e.g., isotrichotriol and trichotriol) in addition to 15-deCAL [9,10,12]. Even if *FgTri3* was disrupted in a trichothecene overproducer, 15-deCAL overproduction was not successful due to the accumulation of earlier bicyclic shunt metabolites [12]. Such metabolite profiles are distinct from those of other trichothecene pathway *Tri* gene mutants, which produce substrates for the knocked-out enzymes and the derived shunt metabolites [1]. For example, an *F. graminearum* disruption mutant of a cytochrome P450 monooxygenase (CYP) gene, *Tri11*, which encodes ITD C-15 hydroxylase (Tri11p), produces ITD and its hydroxylated derivatives, 7-hydroxyisotrichodermin (7-HIT) and 8-hydroxyisotrichodermin (8-HIT), as the main products (Figure 1) [13]. Similarly, a large amount of CAL accumulates in culture by the disruption of *Tri1*, a CYP gene involved in the modification of CAL [25].

A possible explanation for the low production of 15-deCAL by the Δ*Fgtri3* mutant is a functional disadvantage of trichothecene biosynthesis machinery in the absence of Tri3p. It is possible that an interaction with FgTri3p protein may be necessary for the proper assembly and full activity of Tri11p enzyme, which is responsible for the production of 15-deCAL [12]. In this study, we sought to examine this possibility by assessing the 15-deCAL accumulation level of an *F. graminearum* strain in which a structurally intact Tri3p protein (encoded by a catalytically impaired *FgTri3* mutant gene) exists. Since the *FgTri3* catalytic mutant strain still accumulated earlier bicyclic and shunt metabolites as the main products, we further tested alternative approaches to create a pathway to 15-deCAL.

**Figure 1 ijms-25-06414-f001:**
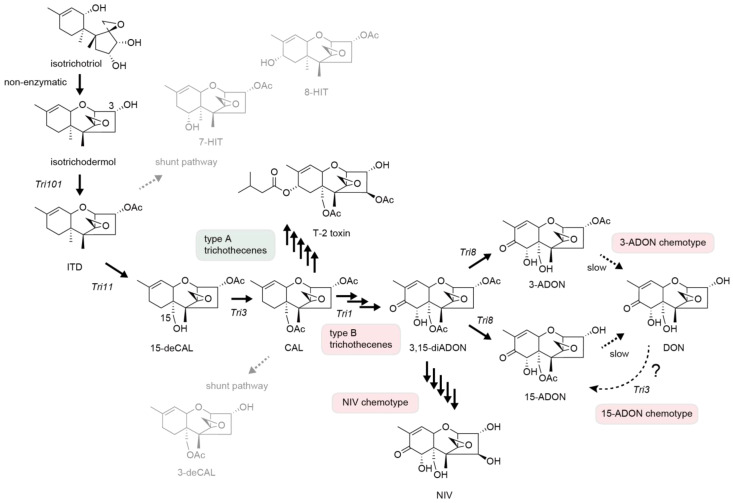
DON biosynthetic pathway of 3-ADON (upper route) and 15-ADON (lower route) chemotypes. The compounds 3-ADON and 15-ADON are relatively stable last intermediates, and thus their co-occurrence often causes a problem. Shunt pathways to 3-deCAL [11], and to 7-HIT and 8-HIT [26], are shown in gray. The Δ*tri11* mutant accumulates ITD and its shunt metabolites, 7-HIT and 8-HIT, in substantial quantities. Although highly specialized knowledge was not provided in the main text, the mutant also accumulates a trace amount of final type B trichothecene products in prolonged culture [13]. The very limited production of type B trichothecenes is likely due to partial complementation of Tri11p function by a microsomal hydroxylase [27], which generally occurs in trichothecene-producing *Fusarium* species. In fact, this CYP inhibitor-insensitive hydroxylase also appears to partially complement the function of Tri11p in *F. sporotrichioides*, as the mutant also produced a trace amount of T-2 toxin [28]. When *FgTri1* is disrupted in the Δ*tri11* mutant background, 7-HIT and 8-HIT are no longer detected from the culture, suggesting that *FgTri1* is responsible for the oxygenation of ITD in the shunt pathway [13]. In contrast to the Δ*tri11* mutants that produce a trace amount of the final trichothecene metabolites along with ITD and its shunt metabolites, the Δ*tri3* mutants of both *F. graminearum* and *F. sporotrichioides* do not produce them: they accumulate a limited amount of 15-deCAL and substantial quantities of earlier bicyclic metabolites [12].

## 2. Results and Discussion

### 2.1. Low Production of 15-deCAL by the ΔFgtri3 Mutant Is Not Caused by a Decrease in Tri11p Activity in the Absence of FgTri3p

Previous X-ray crystallographic studies of *F. sporotrichioides* FsTri3p identified the substrate-binding site of C-15 acetylase, which is located in the tunnel formed at the interface of the N- and C-terminal domains [29]. Most of the interactions with 15-deCAL are hydrophobic, including the binding of carbon C-8 of 15-deCAL to the Cγ of Val-469 (corresponding to Val-467 of FgTri3p). We selected the active site Val-467 to impair catalytic efficiency while maintaining overall FgTri3p protein structure. Ultimately, *FgTri3* was mutated to *FgTri3__V467G_*, in which this Val-467 residue was substituted to a less bulky Gly residue. The resulting *FgTri3__V467G_* strains used in this experiment, EK_001, EK_002, and EK_003 (see Section 3.1), are derived from a transgenic strain YN_149 (Appendix A) that overproduces 3,15-diADON [12]. In these strains, *Tri6* and *Tri10* trichothecene regulatory genes [30,31,32] are overexpressed from the *Aspergillus nidulans* translation elongation factor 1-alpha (*TEF1α*) and glyceraldehyde 3-phosphate dehydrogenase (*GPD*) gene promoters [33], respectively.

Transgenic *Fusarium* strains were inoculated onto liquid YS_60 medium but not onto liquid YG medium in this experiment because non-enzymatic deacetylations of trichothecene side chains are limited with the use of YS_60 compared to the previous result with YG medium [12]. Indeed, the pH of YG medium increased at a later stage of the culture and exceeded over pH 8, under which conditions nonspecific ester hydrolysis proceeds. After 3 days of culture on YS_60 medium, EK_001, EK_002, and EK_003 accumulated large amounts of bicyclic metabolites similar to the case with YN_153, except that a small amount of 3,15-diADON was also detected from these catalytic mutants (Figure 2; Appendix A). The amount of 15-deCAL was limited in the cultures of these *FgTri3* mutants, i.e., YN_153 without FgTri3p, and EK_001, EK_002, and EK_003 with a catalytically impaired FgTri3p__V467G_, both of which have an intact copy of *Tri11*. Thus, the low 15-deCAL accumulation level appeared to be caused mainly by a decreased concentration of Tri11p’s substrate ITD, rather than by the decreased activity of Tri11p due to the disruption of potential protein–protein interactions with FgTri3p.

With limited knowledge about the early-step biosynthesis machinery of trichothecenes, the unsuccessful attempt to increase accumulation of 15-deCAL discouraged us to develop transgenic *F. graminearum* overproducers in the absence of fully active Tri3p enzyme. Inhibition of non-enzymatic second cyclization (resulting in decreased synthesis of ITD) but not of C-15 hydroxylation of ITD is likely to be the main reason that accounts for low accumulation of the reaction product of Tri11p in Δ*Fgtri3* mutants. The present result supports our previous hypothesis in which an active Tri3p enzyme is necessary for releasing the tricyclic precursor 15-deCAL outside of the toxisome membrane. A non-enzymatic second cyclization may proceed by efficient removal of 15-deCAL as the reaction product CAL from the closed-reaction system [12]. To explore the strategy of generating 15-deCAL overproducers with a *FgTri3* deletion, future studies must focus on identifying a specific toxisome efflux pump, if any, which may not be regulated by Tri6p and has a high affinity to 15-deCAL.

### 2.2. Attempts to Establish a Metabolic Route to 15-deCAL with Transgenic Strains Carrying a Native FgTri3 Gene

To design and achieve efficient bioproduction of 15-deCAL in transgenic *F. graminearum* based on currently available knowledge, we attempted to establish a metabolic route to 15-deCAL from CAL by blocking A-ring oxygenation and accelerating C-15 deacetylation. We first investigated the efficiency of inhibiting A-ring oxygenation by culturing the transgenic 3-ADON chemotype (YN_155; Appendix A) on a synthetic methionine medium [34], and then deleted *Tri1* (YN_173; Appendix A) to allow production of CAL by culturing the deletion mutant on more productive YS_60 medium, as follows: (1) *Tri8* was replaced with that of the 3-ADON chemotype (*Tri8__3-ADON chemotype_* encoding a C-15 deacetylase) in the trichothecene overproducer YN_120 [12], yielding strain YN_155, which was induced to produce trichothecenes on a synthetic methionine medium. (2) Disruption of *Tri1* in YN_155 to completely delete its A-ring modification activities, yielding strain YN_173, which was induced to produce trichothecenes on liquid YS_60 medium. However, these approaches did not lead to the accumulation of 15-deCAL as the main product while considerable amounts of other metabolites were observed (Appendix A). In this way, a major route for 15-deCAL production in transgenic strains could not be established, and another strategy based on different experimental approaches was required.

### 2.3. Isolation and Identification of a Fungal Strain That Specifically Deacetylates C-15 of CAL

As an alternative method for a large-scale preparation of 15-deCAL, we have also attempted to obtain 15-deCAL by microbial conversion in parallel experiments. For this purpose, a microorganism capable of specifically deacetylating C-15 of CAL was screened from soil samples. Thirty milligrams of soil samples from diverse geographical regions were suspended in water and serial dilutions of each sample were plated on medium 802 agar. The single colonies that appeared on the plate were examined for C-15 deacetylation activity using a 24-well-culture-plate feeding assay. Among 121 microbial colonies that appeared on the agar plates, a fungal strain of MT-25 isolated from rice field soil in Kusatsu City, Shiga Prefecture, hydrolyzed the 15-*O*-acetyl group of CAL in synthetic 1 × NS medium with a pH of four. The giant colony formed on potato dextrose agar (PDA) was colorless and cottony (Figure 3A), and micro- and macroconidia were induced on CMC liquid medium and carnation leaf agar, respectively (Figure 3B). The phylogenetic position (Figure 3C) was ascertained by aligning the partial sequences of the translation elongation factor1-α gene (*EF1-α*) and the second-largest subunit gene of RNA polymerase II (*RPB2*), which were amplified using polymerase chain reaction (PCR) with previously listed primers [35]. Based on the phylogenetic tree constructed with the combined data of *EF1-α* and *RPB2*, strain MT-25 was positioned in the same group as the *Fusarium commune* strains (NRRL 28387, MRC 2564, and MRC 2566) [36,37].

### 2.4. A Bioconversion Strategy for a Large-Scale Preparation of 15-deCAL

We first generated a CAL overproducer by disrupting *Tri1* in the 3,15-diADON overproducer YN_149 [12], yielding strain YN_171. This strain carries a mutated *Tri8* gene (*Tri8_nsm*) containing a nonsense mutation (see Appendix A) and accumulated a large amount of CAL but not C-3 deacetylated products, such as 3-deacetylcalonectrin (3-deCAL), on liquid YS_60 medium after a prolonged incubation period. This property is advantageous for efficient purification of CAL from a fungal culture.

We next determined which of the following media, YG, YS_60, and 1 × NS, is most suitable for the efficient preparation of 15-deCAL via specific C-15 deacetylation of CAL. CAL was extracted from 100 mL of the YS_60 culture of the CAL overproducer YN_171 with an equal volume of ethyl acetate, and purified by thin-layer chromatography (TLC). For specific C-15 deacetylation, *F. commune* strain MT-25 was inoculated onto 30 mL of liquid media supplemented with a purified CAL sample (corresponding to the amount derived from 30 mL of the YN_171 culture) in a 100 mL Erlenmeyer flask (final 1.0 × 10^4^ conidia/mL) and incubated at 25 °C with reciprocal shaking at 125 strokes/min. While the added CAL was still detected from the YG and YS_60 cultures on TLC, accompanied by the appearance of a more hydrophilic blue spot with an *R*_f_ value equal to that of 15-deCAL at 48 h of incubation (Figure 4A), it completely disappeared from the 1 × NS cultures. LC-MS/MS analysis of the metabolites confirmed the presence of 15-deCAL (Figure 4B). In the 1 × NS culture, CAL disappeared as early as by 36 h of incubation according to TLC analysis, and the intensity of the 15-deCAL spot remained almost constant even on day 7 of incubation. Such a stable bioproduction system permits great flexibility for efficient recovery of 15-deCAL from culture.

### 2.5. Optimization of the 15-deCAL Preparation Method

To optimize the 15-deCAL purification protocol, we investigated the conditions under which CAL is selectively and efficiently recovered from the fermented culture of the CAL-overproducer strain YN_171 (see Appendix A). As a result, we found that 40% the volume of hexane relative to the YN_171 culture is suitable for the recovery of CAL.

To reduce the amounts of contaminating metabolites produced by strain MT-25 during feeding, the conditions of crude CAL feeding to the culture were investigated. From the time-dependent metabolite profiles of the culture fed with two doses of CAL (Appendix A), it was determined that CAL extracted from one volume of the YN_171 culture could reasonably be fed to a half-volume of the MT-25 culture. Up to 4 days of incubation, contaminating ethyl acetate-extractable metabolites with UV absorbance at 254 nm were negligible. From the results, the CAL-fed MT-25 culture was incubated for 4 days to purify 15-deCAL by TLC.

By applying the protocol thus determined (Appendix A), we prepared 15-deCAL; the hexane extract of the CAL-overproducer culture was fed to the *F. commune* MT-25 culture. After extraction with ethyl acetate and purification by TLC, the final yields of the deacetylated product 15-deCAL in duplicate experiments were 4.16 and 4.54 mg, starting from 60 mL of YN_171 culture (average yield of 72.5 mg/L). The yield was much higher than that obtained with cultures of the *Fusarium roseum* wild-type (ATCC 28144) and *F. sporotrichioides* Δ*Fstri3* mutant (MB2972) strains, which afforded 0.075 mg/L [7] and 17 mg/kg [9] of 15-deCAL, respectively.

## 3. Materials and Methods

### 3.1. Strains

All transgenic *F. graminearum* strains used in this study are derived from strain JCM 9873, a 15-ADON producer. Strains EK_001, EK_002, and EK_003 were generated in this study by transforming YN_153 (Δ*Tri6*/P*_TEF_::Tri6*, ΔP*_Tri10_*/P*_GPD_*, Δ*Tri8*/*Tri8_nsm*, Δ*Tri3*/P*_GPD_::hph::tk*), a *Tri3*-knockout strain derived from a trichothecene overproducer [12], with a double crossover homologous recombination vector containing *Tri3__V467G_*, followed by counter selection against the negative marker gene (Appendix A). Strains YN_145 (Δ*Tri6*/P*_TEF_::Tri6*, ΔP*_Tri10_*/P*_GPD_*, Δ*Tri8*/P*_GPD_::hph::tk*), YN_149 (Δ*Tri6*/P*_TEF_::Tri6*, ΔP*_Tri10_*/P*_GPD_*, Δ*Tri8*/*Tri8_nsm*), and YN_153 (Δ*Tri6*/P*_TEF_::Tri6*, ΔP*_Tri10_*/P*_GPD_*, Δ*Tri8*/*Tri8_nsm*, Δ*Tri3*/P*_GPD_::hph::tk*) are pathway *Tri* gene mutants derived from a 15-ADON-overproducing strain YN_120 (Δ*Tri6*/P*_TEF_::Tri6*, ΔP*_Tri10_*/P*_GPD_*) [12]. Strains YN_155 (Δ*Tri6*/P*_TEF_::Tri6*, ΔP*_Tri10_*/P*_GPD_*, Δ*Tri8*/*Tri8__3-ADON chemotype_*) and YN_173 (Δ*Tri6*/P*_TEF_::Tri6*, ΔP*_Tri10_*/P*_GPD_*, Δ*Tri8*/*Tri8__3-ADON chemotype_*, Δ*Tri1*/P*_TUB_::hph::tk*) (Appendix A) were generated in our attempts to establish a metabolic route to 15-deCAL in this study. Strain YN_171 (Δ*Tri6*/P*_TEF_::Tri6*, ΔP*_Tri10_*/P*_GPD_*, Δ*Tri8*/*Tri8_nsm*, Δ*Tri1*/P*_TUB_::hph::tk*) was constructed for the overproduction of CAL necessary for the feeding of *F. commune* MT-25.

### 3.2. Media and Reagents

YS_60 (6% [*w*/*v*] sucrose, 0.1% [*w*/*v*] Bacto™ yeast extract), YG (2% [*w*/*v*] sucrose, 0.5% [*w*/*v*] Bacto™ yeast extract), and synthetic 1 × NS media [44] were used for fungal cultures. Synthetic methionine medium [34] was evaluated for the overproduction of 15-deCAL in strategy (1). For the isolation of microorganisms from soil samples, agar plates of growth medium 802 (1% [*w*/*v*] Hipolypepton, 0.2% [*w*/*v*] Bacto™ yeast extract, 0.1% [*w*/*v*] MgSO_4_·7H_2_O) were used.

TLC plates (Kieselgel F_254_) and LC-MS-grade acetonitrile were obtained from Merck Millipore (Darmstadt, Germany). Th chemical 2′-deoxy-5-fluorouridine (5-FdU) was from Carbosynth Ltd. (Compton, Berkshire, UK). Other materials were purchased from FUJIFILM Wako Pure Chemicals (Osaka, Japan).

### 3.3. Analysis of Trichothecenes

For analysis of metabolites produced by transgenic *F. graminearum* strains, the culture supernatant was extracted with an equal volume of ethyl acetate and the solvent layer was evaporated under nitrogen gas. After reconstitution of the dried materials in a small volume of ethanol, the samples were analyzed by TLC using ethyl acetate/toluene (3:1) as the developing solvent. Trichothecenes on TLC plates were reacted with 4-(4-nitrobenzyl)pyridine (NBP) and visualized by spraying with tetraethylenepentamine (TEPA) [45]. For LC-MS/MS analysis, an Eksigent ekspert™ ultraLC 100-XL (Dublin, CA, USA) connected to an AB SCIEX Triple TOF 4600 System (Framingham, MA, USA) was used. The procedures of the trichothecene analysis followed the conditions described in our previous studies [46].

### 3.4. Purification of CAL and 15-deCAL

After developing the culture extracts on TLC, the edge of the plate was separated from the whole plate. The zones containing the trichothecenes were identified by a blue color development upon reaction with NBP/TEPA. The metabolites in the silica gel layer of the intact plates were carefully scraped off using a spatula and, subsequently, eluted with ethyl acetyl acetate.

## 4. Conclusions

The lack of accumulation of 15-deCAL in a large quantity in the culture of the *Fusarium Tri* gene deletion mutants is noteworthy. This intermediate is the substrate of C-15 acetylase encoded by *Tri3*. The present study showed that the low level of 15-deCAL production by the Δ*Fgtri3* mutant was not due to a functional disadvantage of trichothecene biosynthesis machinery in the absence of structurally intact FgTri3p protein. This supports our previous model, which suggested that a catalytically active FgTri3p protein is essential for efficient non-enzymatic cyclization of earlier bicyclic intermediates. To better understand the possible roles of Tri3p in chemotype diversification, we developed a bioproduction system for 15-deCAL using a transgenic CAL-overproducer strain, combined with the ability of an *F. commune* strain to specifically deacetylate C-15 of CAL, for kinetic studies of Tri3p.

## Figures and Tables

**Figure 2 ijms-25-06414-f002:**
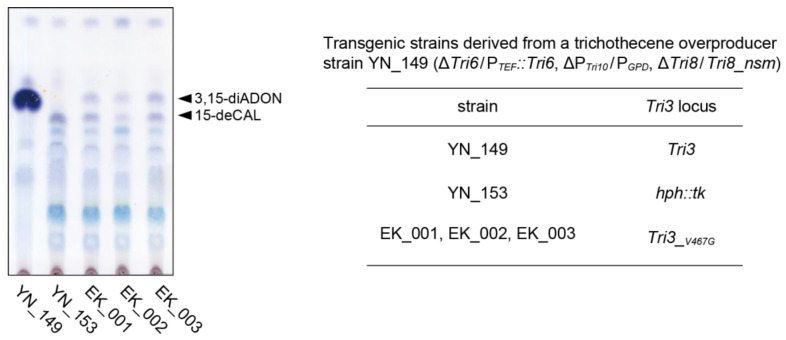
Production of trichothecene intermediate and shunt metabolites by 3,15-diADON-overproducer YN_149, its transformant YN_153 where *Tri3* was replaced with the *hph::tk* maker, and the transformants derived from YN_153 (EK_001, EK_002, and EK_003) where *hph::tk* was replaced by *Tri3__V467G_*. These transgenic strains were cultured on liquid YS_60 medium, and the ethyl acetate extracts of the culture supernatants were analyzed after 3 days of gyratory shaking at 135 rpm and 25 °C. Both strain YN_153 and its *Tri3__V467G_* transformants produced only a small amount of 15-deCAL in addition to the bicyclic metabolites [12]. ITD and its derivatives, 7-HIT and 8-HIT, were not detected from the culture of the *FgTri3__V467G_* mutant (with *Tri* gene overexpression) in this experiment. The existence of catalytically impaired Tri3p in these *Tri3__V467G_* transformants was responsible for the production of a small amount of 3,15-diADON in their cultures. In addition to TLC analysis, the presence and absence of these metabolites were confirmed by LC-MS/MS analysis (Appendix A).

**Figure 3 ijms-25-06414-f003:**
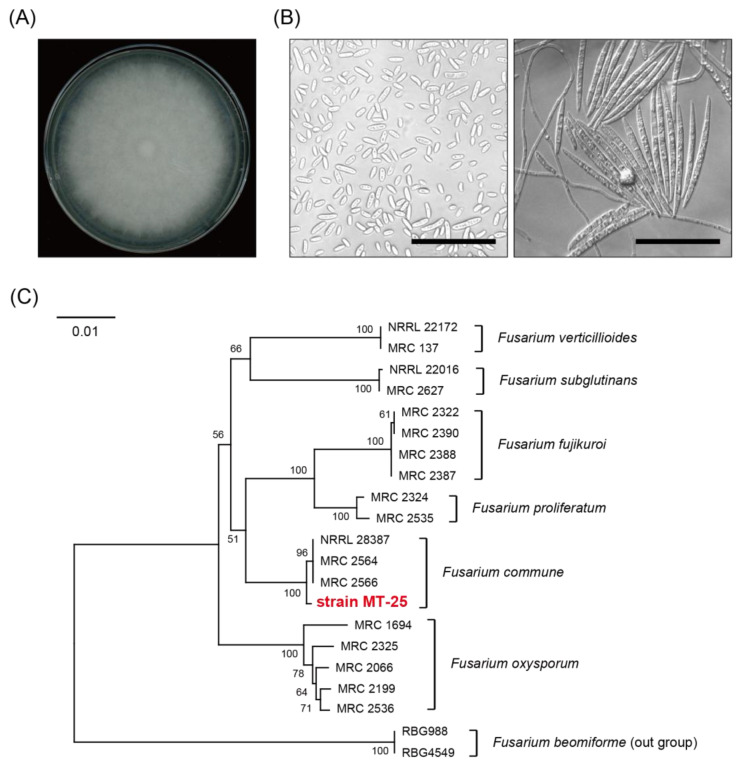
Characterization and identification of *F. commune* MT-25 isolated from rice field soil. (**A**) Top view of PDA plates of *F. commune* MT-25 after 7 days of incubation at 25 °C. (**B**) Micro- (left) and macroconidia (right) induced to form in CMC liquid medium [38] and on carnation leaf agar [39], respectively. Scale bar = 50 μm. (**C**) A neighbor-joining tree of *Fusarium* species based on the combined sequences of the *EF1-α* and *RPB2* datasets [37,40,41,42,43]. Numbers at the node show bootstrap values. Sequences used in the analysis are listed in Appendix A. The scale bar represents the branch length of the mean number of differences per residue.

**Figure 4 ijms-25-06414-f004:**
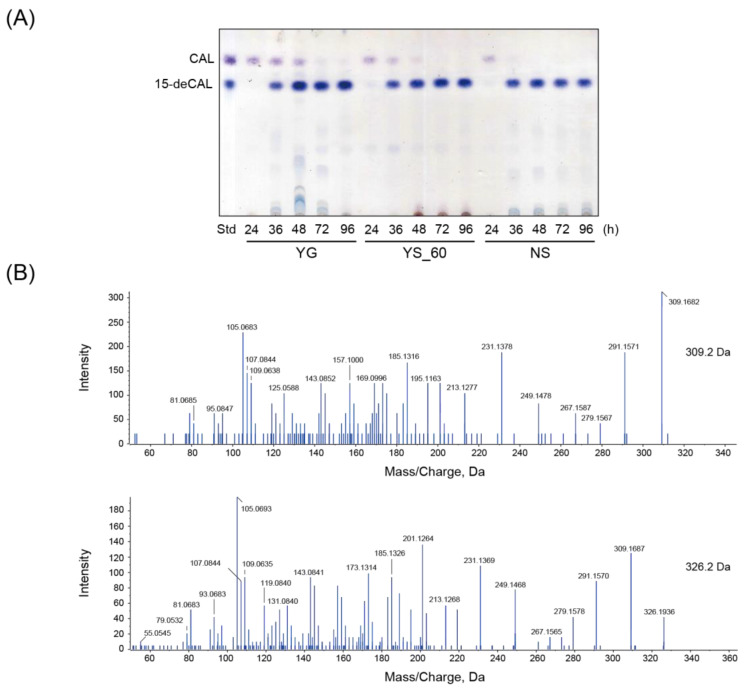
Deacetylation at C-15 of CAL by *F. commune* MT-25. (**A**) TLC of metabolites of strain MT-25 fed with CAL. Strain MT-25 was cultured on liquid YG, YS_60, and 1 × NS media, and the ethyl acetate extracts of the culture supernatant were analyzed at the hours of incubation indicated at the bottom of each sample. (**B**) LC-MS/MS of 15-deCAL in the YG culture supernatant. The tandem mass spectra of precursor ions were detected in positive-ion mode. *m*/*z* 309.2 and *m*/*z* 326.2 correspond to [15-deCAL + H]^+^ (*m*/*z* 309.1697) and [15-deCAL + NH_4_]^+^ (*m*/*z* 326.1962), respectively. The MS/MS spectra were superimposable to those of the 15-deCAL standard [13].

## Data Availability

The original contributions presented in the study are included in the article/supplementary material, further inquiries can be directed to the corresponding authors.

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
