# Peer review of "Attempting to Create a Pathway to 15-Deacetylcalonectrin with Limited Accumulation in Cultures of Fusarium Tri3 Mutants: Insight into Trichothecene Biosynthesis Machinery"

_ijms, 2024, doi:10.3390/ijms25126414_

Round 1

Reviewer 1 Report

Comments and Suggestions for Authors

The objective of the study is not clear to me. It is not clear for what purpose and with what impact the conducted study is intended.

The introduction is redundant and incompleted.

Although the conclusions align with the obtained results, it is not clear what the necessity and impact of the evaluated method are.

Author Response

The objective of the study is not clear to me. It is not clear for what purpose and with what impact the conducted study is intended. The introduction is redundant and incompleted. Although the conclusions align with the obtained results, it is not clear what the necessity and impact of the evaluated method are.

Thank you for the useful comments to improve the manuscript. This paper contributes as a part of molecular and biochemical studies on the biosynthesis mechanism of trichothecenes that have been carried out by the research groups (including the author’s group) for more than 35 years. The pathways and enzymes for DON production were identified and characterized by early 2000s. However, there still remains unsolved mysteries about the function and role of Tri3p in the biosynthesis, which impacts our understandings about the non-enzymatic second cyclization (Lines 76-81) and chemotype diversification (Lines 57-61).

To assess possible involvement of Tri3p in shaping the chemotypes of type B trichothecene producers, it is crucial to characterize the kinetic properties of the C-15 acetylase of different chemotypes in detail; based on the results, we know whether the scenario assuming the involvement of Tri3 in chemotype determination to some extent (i.e., is persistence of 15-ADON in 15-ADON chemotype partially be supported by the 15-ADON chemotype’s Tri3p activity?) is plausible or not (Lines 65-72). In the kinetic experiments, the authentic substrate of Tri3p, 15-deCAL, needs to be used with DON and 3-ADON, but the amount of 15-deCAL that can be obtained from the Δtri3 mutant culture is limited due to the inhibition of the non-enzymatic second cyclization. Thus, the method described in this report contributes to such an experiment.

Another scientific significance of the paper is that the study gives some insight into the unique biosynthetic mechanism of this important mycotoxin. By the results obtained in Figure 2 (2.1. Low Production of 15-deCAL by the ΔFgtri3 Mutant Is Not Caused by a Decrease in the Tri11p Activity in the Absence of FgTri3), the possibility of Tri11p enzyme being fully activated or stabilized by the Tri3p protein through protein-protein (Tri11p-Tri3p_V467G) interaction was unambiguously excluded. Therefore, we were left with the possibility that the C-15 acetylase activity of the Tri3p enzyme, but not the C-15 hydroxylation activity of Tri11p, is the causable factor that affected production of 15-deCAL (i.e., product of the Tri11p enzyme) in the Δtri3 mutant. This result also serves as circumstantial evidence that supports our previous theory that the Tri3p enzyme’s important role in the biosynthesis is moving 15-deCAL out of the toxisome (as the reaction product CAL), thus facilitating the thermodynamically unfavorable non-enzymatic cyclization of isotrichotriol to ITDmol to proceed, as explained by Le Chatelier’s principle (ref. 12).

Ultimately, based on your comments, we have revised Figure 1 and changed the title of the paper from “Finding a Route to the Trichothecene Pathway Intermediate 15-Deacetylcalonectrin with Limited Accumulation in Fusarium Mutant Cultures” to “Attempting to Create a Pathway to 15-Deacetylcalonectrin with Limited Accumulation in Cultures of Fusarium Tri3 Mutants: Insight into the Trichothecene Biosynthesis Machinery” to better represent the purpose and impact of the study.

(Please note that line numbers we mentioned above refer to the revised manuscript and not to the original submission)

Reviewer 2 Report

Comments and Suggestions for Authors

The paper entitled: Finding a Route to the Trichothecene Pathway Intermediate 15-Deacetylcalonectrin with Limited Accumulation in Fusarium Mutant Cultures, by: E. Kasahara, Y. Kitamura, M. Katada, M. Mizuki, N. Okumura, T. Sano, Y. Koizumi, K. Maeda, N. Takahashi-Ando, M. Kimura, and Y. Nakajima, constitutes a new step in the study of the tricyclic metabolites of the Fusarium trichothecene pathway, this time, thea uthors are interested in the  metabolite 15-deCAL, which is the only intermediate that does not accumulate in a large quantity in the culture of the pathway Tri gene mutants. The authors showed that the low level of 15-deCAL production by the ΔFgtri3 mutant was not caused by a functional disadvantage of the Tri11p enzyme in the absence of structurally intact FgTri3p protein. The authors demonstrated that this fact was in support of a previous model from the group demonstrating that a catalytically active FgTri3p protein is essential for efficient supply of the substrate ITD to Tri11p. The authors developed a bioproduction system using a transgenic CAL-overproducer strain, combined with the ability of an F. commune strain that can specifically deacetylate C-15 of CAL, for the production of 15-deCAL, so their bioconversion strategy can be used for a large scale preparation of 15-deCAL. The paper builts in the previous knowledge for the pathway intermediate in the biosynthesis of Fusarium trichothecenes, establishing a method of producing 15-deCAL metabolite by using Fusarium graminearum through the use of a CAL-overproducing transformant, then feeding the crude CAL extract to a Fusarium commune strain that was isolated as capable of specifically deacetylating the C-15 acetyl, and then efficiently recovering 15-deCAL. The paper is well developed and full of details both in the body of the text as well as in the supporting information, and constitutes an advance in the knowledge of the pathways in the biosynthesis of Fusarium trichothecenes.

Author Response

The paper entitled: Finding a Route to the Trichothecene Pathway Intermediate 15-Deacetylcalonectrin with Limited Accumulation in Fusarium Mutant Cultures, by: E. Kasahara, Y. Kitamura, M. Katada, M. Mizuki, N. Okumura, T. Sano, Y. Koizumi, K. Maeda, N. Takahashi-Ando, M. Kimura, and Y. Nakajima, constitutes a new step in the study of the tricyclic metabolites of the Fusarium trichothecene pathway, this time, the authors are interested in the metabolite 15-deCAL, which is the only intermediate that does not accumulate in a large quantity in the culture of the pathway Tri gene mutants. The authors showed that the low level of 15-deCAL production by the ΔFgtri3 mutant was not caused by a functional disadvantage of the Tri11p enzyme in the absence of structurally intact FgTri3p protein. The authors demonstrated that this fact was in support of a previous model from the group demonstrating that a catalytically active FgTri3p protein is essential for efficient supply of the substrate ITD to Tri11p. The authors developed a bioproduction system using a transgenic CAL-overproducer strain, combined with the ability of an F. commune strain that can specifically deacetylate C-15 of CAL, for the production of 15-deCAL, so their bioconversion strategy can be used for a largescale preparation of 15-deCAL. The paper builts in the previous knowledge for the pathway intermediate in the biosynthesis of Fusarium trichothecenes, establishing a method of producing 15-deCAL metabolite by using Fusarium graminearum through the use of a CAL-overproducing transformant, then feeding the crude CAL extract to a Fusarium commune strain that was isolated as capable of specifically deacetylating the C-15 acetyl, and then efficiently recovering 15-deCAL. The paper is well developed and full of details both in the body of the text as well as in the supporting information, and constitutes an advance in the knowledge of the pathways in the biosynthesis of Fusarium trichothecenes.

Thank you for taking your invaluable time for reviewing the manuscript.

Reviewer 3 Report

Comments and Suggestions for Authors

I have just a few suggestions/remarks for you to improve.

Line 65-85: Introduction would benefit if you would explain why, it is important to create 15-deCAL overproducer. What exact experiments/studies would be done with this excreted and collected toxin precursor? Why the reader should care that you are trying to create such mutant strains/production method. You need to convince the reader of the necessity of your work.

Line 102: “which are located” – maybe “is”, you talking about single biding site.

Line 126-127: You cannot claim such bold conclusions. If you failed to make something, it does not prove that it is impossible to do so. Other researches later might find a way. You don’t know how Tri3 interacts with enzymes that are upstream in this pathway. It does not have to interact with Tri11 directly. There may be other signaling proteins involved. You might be right, and maybe Tri3 moving its catalyzed product through membrane and changing concentration gradient is the key. But then somebody might mutate Tri3 in such way that it moves15-deCAL out of toxisome without turning it into CAL. Or maybe introduce a protein that changes CAL back into 15-decCAL later. Maybe, it is not that, but the conformational changes of Tri3 while it moves CAL through membrane activate some other signaling protein that affects earlier stages of this pathway. So maybe, it is possible to create producer with fully functional Tri3.

While we on the same matter, a few sentences could be added to the article about 7-HIT and 8-HIT shunt pathway. Does it involve enzymes or it is spontaneous reaction? Is it possible to create mutants where all the shunt pathways of IDT are disabled so it is forced to accumulate?

Author Response

I have just a few suggestions/remarks for you to improve.

Thank you for the valuable feedback to improve the quality of the manuscript.  

Line 65-85: Introduction would benefit if you would explain why, it is important to create 15-deCAL overproducer. What exact experiments/studies would be done with this excreted and collected toxin precursor? Why the reader should care that you are trying to create such mutant strains/production method. You need to convince the reader of the necessity of your work.

  In the revised manuscript, we explained more clearly that characterization of Tri3p from different chemotypes, especially 3-ADON and 15-ADON chemotypes, is important to understand the role of the C-15 acetylase in chemotype diversification. The reason/impetus for the study (Lines 65-67, 76-81), the hypothesis and approach (Lines 67-72), and subsequent study plan (Lines 72-74) are described in more detail.

In addition, although not mentioned in this manuscript to avoid confusion, our preliminary results obtained from studies on the regulatory mechanism of trichothecene biosynthesis (ref. 32) suggest that the presence of some trichothecene intermediates, such as 15-deCAL, affect regulation of Tri6 expression. 15-deCAL seems to be a key intermediate for the understandings of both biosynthetic and regulatory mechanism of trichothecenes. For these reasons, 15-deCAL is needed in a large quantity for feeding our highly manipulated Tri transgenic strains, and hence, establishing an efficient 15-deCAL production/purification system is necessary.

Line 102: “which are located” – maybe “is”, you talking about single biding site.

 Thank you; following your remark, “are” was changed to “is”.

Line 126-127: You cannot claim such bold conclusions. If you failed to make something, it does not prove that it is impossible to do so. Other researches later might find a way. You don’t know how Tri3 interacts with enzymes that are upstream in this pathway. It does not have to interact with Tri11 directly. There may be other signaling proteins involved. You might be right, and maybe Tri3 moving its catalyzed product through membrane and changing concentration gradient is the key. But then somebody might mutate Tri3 in such way that it moves15-deCAL out of toxisome without turning it into CAL. Or maybe introduce a protein that changes CAL back into 15-decCAL later. Maybe, it is not that, but the conformational changes of Tri3 while it moves CAL through membrane activate some other signaling protein that affects earlier stages of this pathway. So maybe, it is possible to create producer with fully functional Tri3.

 Thank you for the insightful opinions. Yes, you are right. No one can conclusively say that something is impossible. In the revised manuscript, different expressions were used to describe how we have interpretated and dealt with the results (Lines 147-150). We also discussed the possibility of metabolic engineering of creating a 15-deCAL producer (Lines 157-159).

While we on the same matter, a few sentences could be added to the article about 7-HIT and 8-HIT shunt pathway. Does it involve enzymes or it is spontaneous reaction? Is it possible to create mutants where all the shunt pathways of IDT are disabled so it is forced to accumulate?

  First of all, I thank you for the highly specialized and expert comments. I did not mention the detail regarding the results of Tri11 disruption in the main text, as it causes confusion for general researchers working on trichothecenes. In fact, trichothecene biosynthesis is more complex than what is generally described in the available review papers that summarize the pathways and genes from various trichothecene-producing fungi. Although the activity is very weak, Tri11p function is partially complemented by a general microsomal hydroxylase that happens to accept ITD as the substrate in Fusarium species (ref. 27). Even if Tri11 is disrupted, a trace amount of T-2 toxin and 4,15-diANIV/4-ANIV are detected from the prolonged culture of F. sporotrichioides (ref. 28) and F. graminearum (ref. 13), respectively. With regard to 7-HIT and 8-HIT shunt pathway, we have previously shown that FgTri1 is responsible for the hydroxylation of ITD by using a NIV chemotype strain (ref. 13). From the culture metabolites of the Δtri11ΔFgtri1 double gene disruptants, a small amount of CAL is also detected in addition to ITD (ref. 13). The production of CAL is probably due to the activity of the above general microsomal hydroxylase that accepted ITD as the substrate. So as an answer to your question “Is it possible to create mutants where all the shunt pathways of IDT are disabled so it is forced to accumulate?”, I reply that it is not possible to create an ITD producer that does not produce CAL at present. For generation of such a mutant by using the Δtri11 strain as the parent strain through the gene disruption experiment, the microsomal hydroxylase gene needs to be identified and proved not to be an essential gene. We have added such highly specialized information in the legend of Figure 1.

(Please note that line and reference numbers we mentioned above refer to the revised manuscript and not to the original submission)

Round 2

Reviewer 1 Report

Comments and Suggestions for Authors

The authors intend to clarify the trichothecene biosynthesis pathway throw a molecular and biochemical studies. The manuscript has significantly improved with clarification of the impact and the importance to study the trichothecene pathway. However, I believe the manuscript still suffers from a degree of narrowing and writing that is very specific to specialists in this particular topic. It could greatly improve with an attempt at lighter writing in some parts, especially in the abstract, introduction, and conclusion, so that it can be better understood by those who do not work specifically on the subject but might find it interesting for similar or parallel work. Despite being well written, the manuscript presents heavy writing and is difficult to follow.

Author Response

The authors intend to clarify the trichothecene biosynthesis pathway throw a molecular and biochemical studies. The manuscript has significantly improved with clarification of the impact and the importance to study the trichothecene pathway. However, I believe the manuscript still suffers from a degree of narrowing and writing that is very specific to specialists in this particular topic. It could greatly improve with an attempt at lighter writing in some parts, especially in the abstract, introduction, and conclusion, so that it can be better understood by those who do not work specifically on the subject but might find it interesting for similar or parallel work. Despite being well written, the manuscript presents heavy writing and is difficult to follow.

Reply:

Thank you once again for your insightful feedback aimed at enhancing the quality of the manuscript. In light of your suggestions, I endeavored to describe the study's findings and implications in a more general manner, utilizing the term "trichothecene biosynthesis machinery" more frequently, rather than referencing specific genes and metabolites that are not widely known. The manuscript has been revised to be more accessible to readers who may be unfamiliar with trichothecene, yet still interested in the biosynthesis of fungal toxins. In the "Abstract" section, I refrained from using particular names, such as Tri11, isotrichodermin, and 3,15-diacetyldeoxynivalenol, and instead explained the significance of the study in terms of the biosynthesis mechanism and gene evolution. The "Conclusions" section has been revised to convey the objectives and implications of the study more effectively. Lastly, the final paragraph of the "Introduction" section was revised to provide a clearer understanding of the study's objectives. I hope that the improved manuscript will meet your expectations.